# Acute Kidney Injury and Hyponatremia in Ultra-Trail Racing: A Systematic Review

**DOI:** 10.3390/medicina58050569

**Published:** 2022-04-21

**Authors:** Miguel Lecina, Carlos Castellar-Otín, Isaac López-Laval, Luis Carrasco Páez, Francisco Pradas

**Affiliations:** 1Faculty of Health and Sports Sciences, University of Zaragoza, 22002 Huesca, Spain; miglecina@gmail.com; 2ENFYRED Research Group, Faculty of Health and Sports, University of Zaragoza, 22002 Huesca, Spain; franprad@unizar.es; 3Department of Physiatry and Nursery, Section of Physical Education and Sports, Faculty of Health Sciences and Sport, University of Zaragoza, 22002 Huesca, Spain; isaac@unizar.es; 4Department of Physical Education and Sport, University of Seville, 41013 Seville, Spain; lcarrasco@us.es

**Keywords:** ultra-endurance, acute kidney injury incidence, diagnosis, biomarkers, hyponatremia

## Abstract

*Background and objectives*: Ultra-trail races can cause episodes of acute kidney injury (AKI) and exercise-associated hyponatremia (EAH) in healthy subjects without previous renal pathology. This systematic review aims to review the incidence of these two syndromes together and separately taking into account the length and elevation of the ultra-trail race examined. *Materials and Methods:* A systematic review was conducted through electronic search in four electronic databases (PubMed, EBSCO, Web of Science and Alcorze). *Results:* A total of 1127 articles published between January 2006 and December 31, 2021 were included, 28 of which met the inclusion criteria. The studies were categorized according to the length and stages of the race in four categories: medium (42 to 69 km), long (70 to 99 km), extra (>100 km) and multi-stage if they included various stages. A total of 2950 runners (666 females and 2284 males) were extracted from 28 publications. The AKI incidence found was 42.04% (468 cases of 1113), and 195 of 2065 were diagnosed with EAH, accounting for 9.11%. The concurrence of both pathologies together reached 11.84% (27 individuals) from a total of 228 runners with AKI and EAH simultaneously analyzed. Sorted by race category, the AKI+EAH cases were distributed as follows: 18 of 27 in the extra (13.63% and *n* = 132), 4 in the large (5.79% and *n* = 69) and 5 in the medium category (18.15% and *n* = 27). *Conclusions:* According to these results, extra and medium races showed a similar incidence of AKI+EAH. These findings underline the importance of the duration and intensity of the race and may make them responsible for the etiology of these medical conditions. Due to their variable incidence, EAH and AKI are often underdiagnosed, leading to poorer prognosis, increased condition seriousness and hindered treatment. The results of this review urge participants, coaches and race organizers to take measures to improve the early diagnosis and urgent treatment of possible EAH and AKI cases.

## 1. Introduction

Ultra-trails, also known as ultra-endurance races, or simply ultra-races, have recently experienced significant worldwide growth both in the number of competitions and in the number of participants [1,2]. In the USA alone, a total of 546 races were held in 2016, and this number increased to 1073 in 2019. The number of participants also increased from 46,739 to 109,810. This number is only an estimate and could account for just 20–30% of the total in this country [3]. Ultra-trail races take place in unstable and unpredictable environments, which complicates their delimitation, categorization and study [4]. They can be classified based on elevation gain and loss, number of stages, weather, aid stations and total distance, with medium ultra-trails ranging from 42 to 69 km, long ultra-trails from 70 to 99 km and extra-long ultra-trails in the range above 100 km [5]. Research around ultra-trails has sparked great interest in recent years, with a focus mainly on analyzing success with regard to performance-influencing variables [6]. However, due to the increased popularity that these events have been experiencing [7], it is of great interest to understand their consequences on participants’ health not only in the short [8,9] but also in the long term [10]. This could even include analyzing the incidence of medical conditions induced by the races and the type of medical care provided during and after the race [11,12].

The impact of such races on the human body has been investigated from a broad perspective. Studies have focused on describing how ultra-trail races affect the physiological response of different systems, and have analyzed musculoskeletal [13], cardiac [14] and bone damage [15] and even other minor aspects such as chafing [16]. A key aspect to be considered in ultra-trail races is that athletes must be able to self-feed and self-hydrate. In this regard, factors such as experience, sex, pace and environmental temperature can influence the risk of nutrition- and hydration-related mistakes. Inadequate feeding and drinking can lead to increased extracellular water and therefore increased body weight [17]. This occurrence—overhydration—is commonly seen in ultra-races [18,19] and it entails a risk of developing exercise-associated hyponatremia (EAH). EAH is defined by a below-normal serum sodium concentration (<135 mmol/L) during or up to 24 h after physical activity [20]. It has been classified in three grades of severity basing on [Na+]: mild (values from 130 to 135 mmol/L), moderate (values from 125 to 129 mmol/L) and severe (values < 125 mmol/L) [21]. It can present very diverse symptomatology [22], including nausea, dizziness, increased intracranial pressure, loss of consciousness and even death [23]. The incidence of EAH varies depending on the characteristics of the race, with rates ranging from 4 to 51% for ultramarathon runners [24]. EAH has been widely researched, and there is strong evidence that overhydration is the primary cause of its onset in ultra-trail runners [25]. However, hyponatremia may be also euvolic (60% of all cases in hospitalized patients are euvolic) or hypovolemic when it is associated with low plasma volume [26]. Apart from EAH, fluid replacement and fluid loss suffered by ultra-runners during the effort may also result in acute renal impairment [27]. In the Comrades marathon, Boulter et al. found four runners who suffered EAH and acute kidney injury (AKI) together, and all of them were overhydrated and had taken calcium supplements during the race (452 mg) [28]. When ultra-runners suffer EAH due to overhydration, their glomerular filtration rate deteriorates and their ability to dilute urine and excrete free water decreases [29]. Equally, dehydration is another major cause of AKI [30]. AKI is defined as an abrupt (within hours) decrease in glomerular filtration, with a consequent increase in the concentration of nitrogenated products in the blood, with or without associated oliguria [31], which encompasses both injury (structural damage) and impairment (loss of function) [32]. Despite the different classifications—RIFLE (Risk, Injury, Failure, Loss, End-stage), AKIN (Acute Kidney Injury Network) or KDIGO (Kidney Disease: Improving Global Outcomes)—AKI is always diagnosed when comparing serum creatinine (SCR) changes (a 1.5 fold increase in serum creatinine or >25% decrease in glomerular filtration (eGFR) rate for RISK stage) or urine output (<0.5 mL/kg/h for 6 h). However, the stage and severity of AKI varies according to the criteria used [33].

AKI includes the possibility structural injury such as a decreased kidney function [32], which could be associated with a significant increase in morbidity and mortality in both the short and the long term [34]. Episodes of combined AKI and EAH have been described in hospitalized patients with frequencies ranging from 5% to 42% and varying degrees of severity [35,36]. Nevertheless, no data have evaluated the presence of AKI+EAH together in the sports field, let alone in ultra-trail races.

Considering the nature of these races and their growing popularity, it seems necessary to gather information on their impact on runners’ bodies. Thus, the main objective of this systematic review was to determine AKI and EAH incidence individually and the concurrence of both medical conditions, studying their incidence in the different race categories and finally evaluating their effect and severity in the ultra-trail context.

## 2. Materials and Methods

### 2.1. Criteria for Study Search and Selection

A systematic search of the relevant literature was conducted following the PRISMA method [37]. Selected articles were related to AKI and EAH in ultra-trails and published in the last fifteen years, with a deadline of 31 December 2021. A structured search was performed in different sources of high-quality information in the field of health and sport sciences using the following information databases: PubMed, EBSCO, Web of Science and Alcorze. The search terms and keywords included a mix of medical subject headings (MeSH) and some free-text words for ultra-trail sport. The keywords used in the next equation of the search were: “ultra-endurance,” OR “ultra-trail,” OR “ultramarathon, “OR “off road race,” AND “kidney damage,” OR “renal impairment,” OR “acute kidney injury,” OR “renal injury,” OR “renal damage biomarkers,” AND “hyponatremia,” OR “exercise associated hyponatremia,” OR “electrolyte disorder,” OR “sodium concentration,”. In the Appendix A section, all the equations used are included. 

### 2.2. Inclusion Criteria

Case studies, observational studies, longitudinal or prospective studies.Adult population of both sexes.Races longer than 42 km with one or more stages, held in a natural setting.Analysis of biomarkers related to the prevalence of AKI.Analysis of biomarkers related to EAH.Articles written in English or Spanish.

### 2.3. Exclusion Criteria

Patients with previous chronic renal pathology.AKI or EAH was not assessed.Combined sports (triathlon or biathlon), short duration (<42 km) or track-like environments without elevation gain (>500 m).

### 2.4. Data Extraction

Two authors identified papers through database searching (M.L. and F.P.). During the review process, the following information was collected from each of the studies: year of publication; author; participant-related data (number, mean age and sex); type of event (distance, elevation gain and number of stages); environmental conditions (temperature and degree of relative humidity); body weight loss; pre and post serum creatinine (SCR); pre and post estimated glomerular filtration rate (eGFR); AKI criterion used; RIFLE (Risk, Injury, Failure, Loss, End-stage) kidney disease classification, AKIN (Acute Kidney Injury Network), or KDIGO (Kidney Disease: Improving Global Outcomes); degree and severity of AKI, pre and post EAH ([Na^+^] values); EAH severity. Once the inclusion criteria and exclusion were applied to each study, the aforementioned data were extracted independently by two authors (M.L. and F.P.) using a spreadsheet (Microsoft Inc, Seattle, WA, USA). Disagreements were resolved by discussion until a consensus was reached. 

### 2.5. Overall Quality of Included Studies

Included papers were selected by two independent (M.L. and L.C.) reviewers according to the established inclusion and exclusion criteria. Mendeley Desktop^®^v.1.18.9 (Elsevier, Amsterdam, The Netherlands) was used to remove duplicate articles and analyze the titles and abstracts. When necessary, a further full-text analysis was conducted. Decisions were always approved by both reviewers. However, in case of doubt or discrepancy, a third reviewer was consulted to solve the disagreement. The research and analysis process lasted a total of three weeks. The procedure followed in the review is described in detail in the flowchart (Figure 1).

### 2.6. Risk of Bias Assessment and Methodological Quality

Episodes of AKI and EAH have been evaluated in prospective descriptive, cohort and even comparative studies, but no articles were found that met the inclusion criteria with trial or intervention-type designs in ultra-trail races. Descriptive studies are not assessable with methodologically validated tools such as PEDro [39] or Cochrane Collaboration’s tool [40]. The tools used for the quality assessment of observational descriptive studies specify that their use should be solely illustrative and should only offer qualitative criteria [41]. Regardless, it was decided that the degree of evidence of the included studies should be evaluated using two tools. Firstly, the Quality Assessment Tool for Observational Cohort and Cross-Sectional Studies of the National Heart, Lung and Blood institute [42] was used. According to these specific tools, two authors (L.C and I.L) independently assessed the risk of bias in the included studies. This scale has 14 items that can be responded to with yes, no, Na (not applicable), Nr (not reported) and Cd (cannot be determined), and the scale rates studies as good, fair or poor. All 14 items were evaluated for each article by two different authors (L.C and I.L). Inter-reviewer disagreements were resolved by consensus. Arbitration by a third reviewer was used for unresolved disagreements.

The tool does not use the number of “yes” answers to establish each of the categories but rather leaves the decision of defining cutoff values for each category up to the user. Articles were classified as good when 11 or more items were marked “yes”, fair for 7 to 10 items and poor when only 6 or fewer items were marked “yes”. Secondly, the Rosenbrand et al. [41] tool for level of evidence was used, where studies are evaluated with four letters (A, B, C, D). 

## 3. Results

A total of 1127 potentially relevant studies were identified for the review (Figure 1). Forty more articles were added after their selection from reference lists in other studies, identified through other media. After reviewing titles and abstracts, duplicates were excluded, and the number of articles was thinned out to 219. These selected articles were fully reviewed and evaluated to verify whether they met the inclusion criteria. A total of 28 articles were accepted for final review, 13 of which focused on AKI (Table 1) [28,43,44,45,46,47,48,49,50,51,52,53,54] and the remaining 15 on EAH (Table 2) [24,27,55,56,57,58,59,60,61,62,63,64,65,66,67]. Nine of the studies reported combined results of AKI and EAH [24,27,28,44,45,49,51,52,58]. The distribution of AKI cases in the different categories of ultra-trail races are fully shown in Table 3. EAH incidence sorted by race categories are expressed in (Table 4). The concurrence of AKI+EAH are summarized in (Table 5).

The results of the review are shown in Table 1, Table 2, Table 3, Table 4 and Table 5. For better understanding, the results are presented in 5 sections. The first, Section 3.1, describes the types of races and the sample characteristics of the different studies. Section 3.2 describes the cases of AKI and the markers used for its diagnosis. Section 3.3 describes the cases of EAH. Section 3.4 presents the cases of combined EAH and AKI. Section 3.5 assesses the methodological quality of the studies included in the systematic review.

### 3.1. Types of Ultra-Trail Races and Participant Data

A total of 30 different ultra-trail races were analyzed in 28 studies, all of which were observational with the exception of two comparative studies [50,66]. The study by Martinez Navarro et al. compared two ultra-trail races of different distances [50]. On the other hand, Knechtle et al. evaluated a stage race and a 100-plus km race, in addition to other ultra-endurance sports, such as triathlon or cycling, which were not considered in the AKI and EAH data review as they did not meet the inclusion criteria [60]. Races were divided into four major categories according to the length and the existence or not of stages: twenty studies were single-stage and four were multi-stage. The latter category included two 250 km races with six stages [43,49], one 225 km race with five stages [63] and one 350 km race with seven stages [65]. The one-stage category included races ranging from 45 to 217 km. Single-stage races were classified into three different groups according to length [5]: 14 extra-long races (≥100 km) [24,44,47,50,51,52,53,54,57,58,59,62,64,66,67], 3 long ones (70–99 km) [27,28,45] and 8 medium ones (>42.195–69 km) [44,47,49,54,55,60,64,65]. The cumulative elevation gains and losses ranged between 645 m [59] and 12,000 m [51]. The population sample included 2284 men (77.42%) and 666 women (22.57%), which matches the results from other studies that assess participation by sex in this type of race [68,69]. The average age was 42.01 ± 2.95 years, also similar to the data shown in other similar studies [7,70].

### 3.2. AKI

The total number of AKI cases found in the present systematic review was 468 and the incidence was 42.04% (*n* = 1113). AKI cases were classified according to severity using the AKIN, KDIGO or RIFLE criteria [31,33]. AKI cases were distributed as follows: 392 subjects met the risk-stage criteria (35.22%) [16,44,45,46,47,49,50,52,54], 75 subjects suffered AKI in the injury stage (6.73%) [24,28,42,46,48,53] and only one subject met the criteria for failure (0.01%), but he did not require renal replacement therapy. This case occurred in a 95-mile ultra-trail [24]. Five of the reviewed studies evaluating AKI found no cases [27,47,52,58,67].

A total of 107 cases (67,72%) were found in the multi-stage races (*n* = 158) [43,49], of which 70 were classified in the risk stage and 37 in the injury stage. No runner reached failure-stage AKI. In the single-stage races, 361 cases of AKI were found (*n* = 955): 322 in the extra-long races (*n* = 826) [24,43,46,49,51,53], five in long races (*n* = 69) [28,46] and only 34 in the medium-distance races (*n* = 60) [45,50]. All AKI cases categorized by severity and race category are fully shown in Table 4. Two biomarkers were used for AKI diagnosis: pre and post-race SCR level, used in 14 of the 15 studies that assessed AKI [28,43,44,45,46,47,48,49,50,51,53], and eGFR through the use of SCR-derived equations [44,45,46,50,51,52,54]. One of the highlights was the innovative method based on the analysis of Cystatin C (Cys C) that was used in one of the studies [44]. Previous data reveal a greater use of the SCR biomarker compared to the eGFR one, despite the latter being considered the gold standard in clinical practice. The values of both markers were collected in Table 1 and Table 2 (pre-race and post-race), when available. Mean pre-race SCR values ranged between 0.72–1 mg/L. Post-race SCR values multiplied between 1.03 [44] and 1.71 [45] when compared to pre-race values. eGFR decreases ranging from 29% [44] to 47.7% [45] were reported in all studies.

### 3.3. EAH 

A total of 195 cases of EAH were found (*n* = 2065) with an incidence of 9.44%, of which 172 (8.33%) were diagnosed as mild [24,28,44,46,49,51,54,57,61,62,63,64,66] and 23 (1.11%) as moderate [27,49,51,56,66]. No runners were reported as having symptoms of severe EAH. In addition, no cases of hospitalization or medical treatment were found. Seven of the analyzed studies did not report any cases of EAH [45,52,55,58,60,64,65]. Fifteen mild cases were found among the multi-stage races (*n* = 194) [63,65]. Single-stage extra-long races made up 157 out of the total 195 registered cases [24,49,51,56,57,61,63]. A total of six cases were reported among the long category races (*n* = 69) [27,28], and 13 cases among the medium category ones (*n* = 471) [44,49,54]. Races with no EAH cases were found in all categories. All EAH cases categorized by severity and race category are fully shown in Table 5.

### 3.4. Concurrence of AKI and EAH 

Nine studies evaluated simultaneous AKI and EAH [24,27,28,44,45,49,51,52,58]. In 27 cases, the subjects suffered both pathologies (12.98%). Of the total ultra-trail runners included in this review, EAH and AKI were studied (*n* = 228). Of the 11 studies whose main objective was AKI, 6 studies also evaluated EAH episodes [28,44,45,49,52,67]. A total of 18 cases of AKI+EAH (13.63%) were found in extra-long races (*n* = 132) [24,49,51,52], 4 subjects suffered both AKI and EAH (5.79%) in long ones (*n* = 69) [28] and 5 (18.51%) cases were reported in medium races (*n* = 27) [45,50].

### 3.5. Level of Evidence from Included Individual Studies

The descriptive design of the included studies limited the use of standardized and validated tools, as indicated in Section 2.4. However, the level of evidence of each study was assessed using the tool of Rosenbrand et al. [41] and the National Heart and Lung and Blood Institute [42]. Results are shown in Table 6.

## 4. Discussion

The presence of AKI and EAH in ultra-endurance sports has been studied in previous systematic reviews that separately analyzed both markers [30,34]. However, this systematic review jointly addresses AKI and EAH incidence specifically in ultra-trail races and analyzes the concurrence of AKI+EAH. Consequently, three sections are included: in the first one, AKI and its risk factors and diagnostic biomarkers are assessed; the second section assesses EAH; the last section describes the simultaneous development of AKI and EAH in ultra-races.

### 4.1. AKI 

The main incidence of AKI episodes found in this systematic review was 42.04%, and 468 cases (*n* = 1113). Considering the category of the races, a great heterogeneity exists, with AKI incidence ranging from 7.24% in the long races to 56.66% in medium races. These results are similar to those described in other systematic reviews [30,34]. Despite the high incidence, only one case met the failure stage, which reinforces previous findings suggesting that the alterations suffered by runners are transitory and resolve days after with no medical intervention required [53]. Hernandez et al. analyzed in a case study the possibility of completing a 62 km race after two kidney transplants. No alteration was found in either biomarker or kidney function [71]. It has been hypothesized that longer distances may increase the AKI incidence; however, in our study, medium races showed higher values than longer races. This finding underlines the importance of running speed as an important factor for generating AKI. In moderate exercise, renal flow may fall to 25%; however, in extreme exercise, eGFR may decrease by up to 50% [34]. 

Nevertheless, it should be highlighted that AKI can be challenging to diagnose in its early stages and that it is often syndromic and asymptomatic [72]. AKI is diagnosed through SCR, eGFR and urine analysis. The most widely used biomarker was SCR, which was used in 14 studies [24,27,28,43,44,45,46,47,48,49,50,51,59]. The eGFR method was only used in six studies [44,45,46,50,51,52,54], despite being considered the gold standard for AKI diagnosis [73]. In the study conducted by Wołyniecet et al. during a 100 km track race, it was found that eGFR did not vary, while SCR increased by more than 24.53% [74]. In this review, some studies that assessed both markers showed opposed diagnoses. For instance, some subjects who met AKI criteria according to SCR levels did not meet the criteria when using the eGFR marker [50]. Furthermore, other subjects did meet the criteria according to eGFR yet did not when assessing SCR [44]. Two studies were found in which both markers reflected the same diagnosis [46,50]. Though it is obvious that athletes will experience a rise in SCR at some point during the race [49,75], and despite the existing evidence, the real impact of high SCR on kidney injury remains unclear. It is also unknown whether it constitutes a real pathological entity or whether it is just a sign of the body’s stress response [44].

Multi-stage races show higher declines in eGFR values compared to single-stage races. In multi-stage races, eGFR stabilizes at 24–72 h, with drops in SCR and eGFR observed in rest days between stages. This recovery is linked to rehydration, kidney recovery and the slower running pace [76] that is common in multi-stage races [77]. Conversely, data collected in this review showed higher rates of AKI in multi-stage races [43,49] than in single-stage ones. This could be explained by the extreme heat in which the former took place, given that electrolyte and body water loss can lead to EAH, dehydration and eventually AKI. Both races studying AKI were held in a desert [43,49]; temperatures were thus extremely hot during the day and cold at night. Another factor that has been proposed for explaining the high incidence of AKI in multi-stage races is the running pace at which these races are completed; the recovery between stages allows athletes to maintain a faster race pace than in large races, where they have to complete longer distances in a row [78,79]. 

Regarding single-stage races, Martínez Navarro et al. [50] investigated the incidence of AKI based on race distance in two ultra-trail races: one 107 km-long one (*n* = 43) and another 65 km one (*n* = 19). The results only showed significant SCR level increases in the 107 km race, with no differences in AKI incidence between the two races either immediately post-race (65 km: 44%; 107 km: 56%; *p* = 0.42), or 24 h later (65 km: 7%; 107 km: 0%; *p* = 0.1). Belli et al. [51] considered elevation gain to be another interesting variable but found no cases of AKI after analyzing a 217 km-long race with 12,200 m of cumulative elevation gain. However, an observational study by Hoffmann et al. [47] found that 36% of participants developed risk of AKI (*n* = 227) and 4.9% were injured (*n* = 36) in a 161 km race with a cumulative elevation gain of 12,500 m. It therefore seems clear that the elevation gain and distance are not the only factors that can influence AKI development.

Hoffman et al. [47] studied AKI recurrence in 627 runners who participated in a 161 km race. The sample was divided according to whether athletes had previously suffered AKI episodes in other ultra-trails. Three groups were created following the RIFLE criterion for SCR: no risk, risk and injury. Runners who had previously reached a state of failure were not included. Post-race SCR values were higher in the injury group than in the risk group (1.46 vs. 1.98). Additionally, 227 cases of risk-stage AKI and 31 cases of the most severe stage were found. By contrast, Poussel et al. [44] only found one among 24 study subjects who was diagnosed with the RIFLE risk stage in a 120 km-long race. Although temperatures were moderate (8,6–11 °C), the humidity was extreme (89–99%) in this race, which may be why so few AKI episodes occurred. Similarly, Cairns et al. [52] used the RIFLE diagnostic criterion and found no AKI cases in a sample of 15 runners who participated in a 100 km race with lower humidity and milder temperatures (17.3–21 °C) than the previous case.

### 4.2. EAH

Cases of EAH were found in 195 athletes (9.44%) of a sample of 2065 runners, fewer than in other similar studies [22,80]. The EAH incidence according to race category was more homogeneous, ranging from 2.69% of 482 subjects in medium-distance races to 12.19% in extra ones (*n* = 1209). Research shows that factors associated with EAH are: exercise duration [36], slow running pace [81], female sex [21], low body weight [82], excessive pre-race hydration, hydration protocols using hypotonic drinks according to athletes’ experience [83], NSAID use [84] and extreme heat or cold conditions [63,85]. The results here obtained contrary to AKI incidence seem to support the idea that longer races seem to increase the risk of suffering EAH.

Types of ultra-trail races and development of EAH

As was indicated above, exercise duration is considered another important factor when it comes to EAH [18,21,85]. Several papers have explored whether race duration increases the frequency and severity of EAH. In medium-distance races, some mild EAH cases were found [44,49,54], but no moderate ones were detected. As for long-distance races, Boulter et al. [28] found four (100%) mild EAH cases, while Jouffroy et al. [46] found none. A higher number of cases were found among extra-long races. In one of the most relevant papers, Hoffman et al. [57] found 88 cases of mild EAH and 13 cases of moderate EAH among the total 669 subjects. Similar results were described by Winger et al. [58], who found 12 cases of EAH in a sample of 207 participants. Three other studies also reported EAH cases, with incidences of seven, eight and four cases, respectively [24,61,63]. By contrast, Bracher et al. [59] found no participants suffering from EAH. As for the multi-stage races, Knechtle et al. [65] identified 7 mild cases among 120 participants and Costa et al. [63] found 8 mild cases among 74 participants. 

Hydration and body weight

Body weight and excessive consumption of beverages with or without [Na^+^] supplementation are two relevant factors for developing EAH [17,82,86]. However, there is still no consensus in the ultra-trail hydration guidelines regarding how much liquid should be consumed or how much body weight athletes should loose during the race [21,83]. BDWL seems to be an effective measure for preventing EAH. However, neither the exact percentage of weight that runners could lose during the race nor the body composition measurement methodology are fully agreed on for ultra-races [86]. Cejka et al. [64] suggested an alternative to BDWL for the measurement of overhydration, based on the increase in runners’ arm and leg volume. They proved that an increase in liquid consumption had a direct effect on runners’ feet volume (*r* = 0.54, *p* < 0.0001), and was inversely related to the [Na+] decrease (*r* = −0.28, *p* = 0.0142) and to EAH. However, this method has not been standardized and BDWL continues to be used in research on a regular basis.

Though some guidelines recommend a 2% BDWL [21], several papers in this review reported cases of EAH with equal or even higher BDWL values. This may occur as a result of efficiency in runner’s metabolic pathways, achieved through adaptation to training. By obtaining more water during the oxidation of energy substrates, they produce greater amounts of endogenous water [87]. These physiological adaptations should be helpful for the avoidance of EAH risk. However, most guidelines indicate that dehydration begins at the 3% BDWL mark. In fact, dehydration, extreme sweating and BDWL are all cited as causes of EAH. Thus, excess BDWL could be directly linked to dehydration. In the absence of another more reliable measure, BDWL is still the most widely used in research. Several studies found weight loss of less than 2% in runners who did not suffer from EAH [27,28,49,55,56,59,62,63].

Environmental race *conditions*

Environmental conditions have also been studied as a risk factor associated with the occurrence of EAH in long mountain races, especially when temperatures are extremely cold or warm [23,88,89]. Chlibkoval et al. [56] found no cases of EAH in an observational study conducted in extreme cold (−7.96 to −20 °C), but with a small elevation gain (764 m). In addition, participants barely lost body weight (−0.8 kg). These results contradict most research suggesting that a weight loss of at least 2% during the race is necessary to avoid EAH [21]. Similarly, Costa et al. [63] found 8 mild EAH cases among 74 runners who participated in a 225 km-long stage race with high temperatures (32–40 °C). The average weight loss was only 1.2%.

### 4.3. AKI and EAH

The concurrence of AKI+EAH (11.87% of 228 subjects) was significantly higher in medium races and in extra-long ones (18.51% vs. 13.63%) than in the larger races (5.79%). These results fall in line with other previous studies that found AKI+EAH incidence ranging from 5–42% [35,36]; however, these studies took place in a clinical setting. The incidence found is relevant and supports the idea that the etiology of these pathologies could be related to their appearance. Three main factors increase the risk of developing AKI in ultra-trail races: rhabdomyolysis [30,76,90] NSAID use [84] and EAH [36]. The simultaneous onset of EAH and AKI has been less studied than the rhabdomyolysis and AKI combination, which has produced several publications and even a systematic review [30]. This may be because the relationship between AKI and EAH can seem vague. However, the simultaneous appearance of these two conditions is quite common, according to several studies [24,28,44,49,51].

#### 4.3.1. Overhydration and Rhabdomyolysis

The pathophysiological mechanism of EAH is closely linked to excessive hypotonic fluid consumption and weight loss. Other key factors are sex—with women being at higher risk than men—and lack of racing experience [53], which can lead to overhydration and EAH. On the other hand, excessive fluid intake combined with pathophysiological defects such as the inability to suppress antidiuretic hormone activity [19] or defects in the mobilization of intracellular [Na^+^] into the blood stream could favor the accumulation of intracellular water. This intracellular fluid increase would generate weakness in the myocyte cell membranes and promote their rupture and rhabdomyolysis, releasing myoglobin and other elements into the bloodstream [88]. To our knowledge, only one study focused on the concurrence of AKI and EAH. Cairns et al. [52], in a comparative study, evaluated the relationship between pre-race EAH status and eGFR values. Only runners with moderate EAH showed larger decreases in eGFR, in the five measurement points during the 174 km race and even 24 h after the end of the race. These data prove that higher [Na^+^] concentrations produce larger eGFR reductions. However, none of the athletes met AKI criteria despite the reductions in eGFR. Therefore, it is impossible to state that EAH causes AKI. It does, nonetheless, cause a decline in eGFR.

#### 4.3.2. Race Elevation Gain and Increased AKI and EAH

Another factor that has been highlighted to explain the association between EAH and AKI through rhabdomyolysis in these races is the large cumulative elevation gains [89]. Race mechanics involving a major eccentric component of muscle contraction could result in rhabdomyolysis through the increase in CK and other inflammation markers and muscle destruction. Laboratory ergometer studies have shown that the eccentric component in descents produces greater motor unit recruitment. Muscle fiber rupture stemming from these situations is assessable through rises in CK [90]. However, CK increases were not seen in all the analyzed ultra-trail races, the values were very variable, and the peaks were found at different moments during the races, even in very flat events. The absence of muscular damage markers and rhabdomyolysis could be due to a slower race pace, which would compensate for the eccentric component of steep descents [89]. An example which illustrates that greater elevation gains/losses do not imply greater increases in SCR or CK is the observational study by Belli et al. [51], carried out in a single-stage 217 km race and with an elevation gain of 12,200 m. Rises in CK were found and AKI, SCR and eGFR markers were altered, but no athletes suffered AKI. In a race with only 645 m of elevation gain, Bracher et al. [63] obtained very similar results in SCR and [Na^+^] reduction but did not meet the criteria for AKI or EAH. The explanation why hillier races do not necessarily involve greater CK or SCR rises than flatter ones lies in speed. Despite the greater muscle rupture and motor unit recruitment occurring in the more mountainous races, the slower pace maintained in these events allows for the avoidance of higher concentrations of CK and SCR. Martínez Navarro et al. [50] examined two ultra-trail races with similar elevation gains but different distances. They found contradictory results, with higher SCR levels in the athletes from the 107 km race, and higher EAH in athletes who ran the 65 km race. This might indicate that the pace is more linked to EAH than to AKI. A very recent investigation showed an association between CK values and EAH. One study compared runners with and without exercise-associated hyponatremia in the same 100-mile ultra-marathon. Runners with exercise-associated hyponatremia had less experience over 100 miles and a higher CK after completing the race and even during the recovery period [58].

##### Limitations

The present review excluded studies that were not in Spanish or English. Therefore, a language bias might exist. Another limitation in this review stems from the very nature of AKI. The different diagnostic criteria (AKIN, RIFLE and KDIGO) and different biomarkers (SCR, eGFR and urinary volume) complicated the homogenization of results and made a meta-analysis impossible. The lack of this type of analysis significantly limits the drawing of conclusions with a greater degree of evidence. As for study design, the largest bias lay in the absence of intervention-type randomized studies with control groups. However, the studies were evaluated with two different tools to ensure individual quality and degree of evidence quality, increasing the external validity of the systematic review. Lastly, the variety of ultra-trail races and the terminological confusion regarding their categorization and definition poses another significant risk of bias. Although the inclusion criteria set a lower limit of 42.195 km and natural environments, there was still a wide range of distances among the races, as shown in Table 1 and Table 2.

## 5. Conclusions

Ultra-trail races might cause AKI and EAH, but the incidence is low in healthy individuals without previous pathology. These syndromes often go unnoticed and naturally re-solve without complications. However, the severity and potential repercussions on runners’ health should lead participants to focus on preparation. In addition, raising awareness about the severity of this type of condition among organizers, coaches and medical staff is important. Hydration, NSAID consumption, race pace, runners’ morpho functional characteristics, sex and route details are determining factors in the development of both conditions. Certain pathophysiological mechanisms common to AKI and EAH might favor their simultaneous development, with rhabdomyolysis and overhydration playing a key role in this connection. New studies on ultra-trail races in which EAH markers are systematically assessed should be conducted. Additionally, there is a need for tools that allow for precise AKI diagnosis and biomarkers that enable early detection of kidney damage. Current guidelines and recommendations serve as references for races and participants, but they cannot be considered standards due to the great heterogeneity that exists in the field.

## Figures and Tables

**Figure 1 medicina-58-00569-f001:**
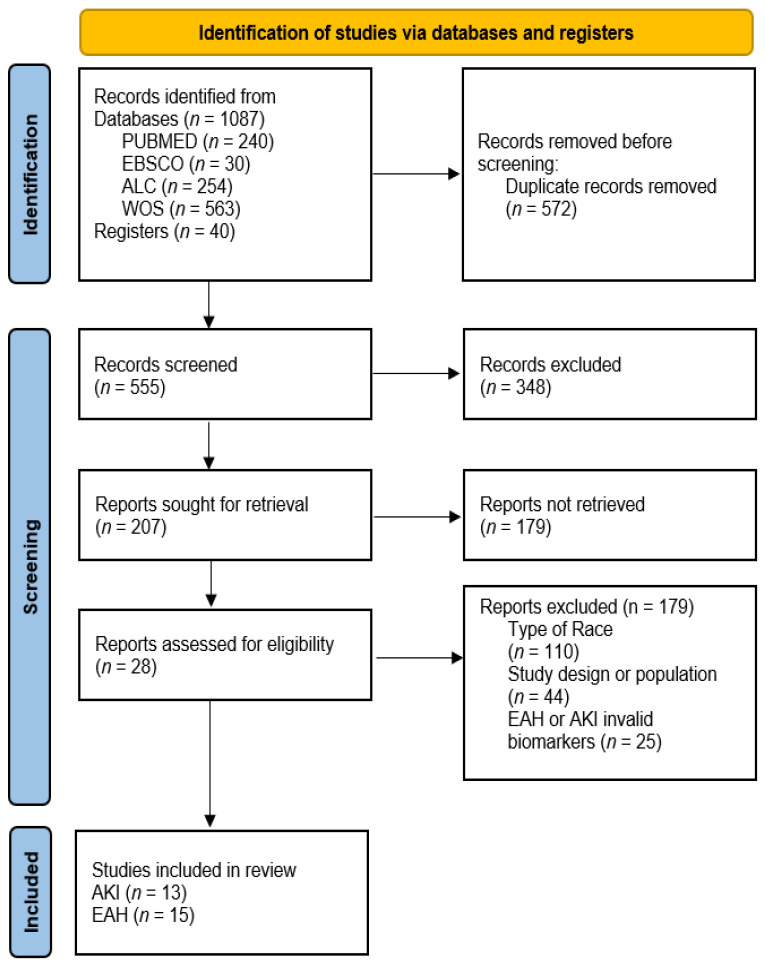
Flow chart describing the selection process of the included studies [38].

**Table 1 medicina-58-00569-t001:** Included studies examining AKI as a primary variable in subjects participating in ultra-trail races.

Author; Year; Reference	Population; Number; Sex; Age	Race; Length, Stages, Elevation	Temperature (Range); Humidity (% Relative)	AKI Biomarkers Pre, Post and Rec SCR (mg/dL) eGFR (mL/min/m²)	EAH Biomarkers [Na+] (mmol/L); Body Weight Loss	AKI Criteria (Stage; *n*; %) EAH Criteria (Stage; *n*; %)
Lipman, 2014 [43]	30;	250 km;	-	↑SCR	-	RIFLE;
	7 F, 23 M;	6;		Pre 1 ± 0.25;		Risk (8, 57%)
	39.6 ± 10 yrs	-		Post 1.4 ± 1.3		Injury (1, 7%)
Poussel, 2020 [44]	24; 1 F, 23 M; 36.5 yrs	120 km; 1; 5700+ m	8.6–11.1 °C; 89–99%	↑SCR Pre 8.6 Post 8.9 ↓ eGFR Post (−29.2%)	-	KDIGO; Risk (1, 4.2%) Injury (0, 0%)
Hoppel, 2019 [45]	8; 0 F, 8 M; 41.5 yrs	67 km; 1; 4500+ m	17–37 °C; 54%	↑SCR Pre 0.9 ± 0.05 Post 1.54 ± 0.29 Rec 1.03 ± 0.14 ↓ eGFR Pre 106.23 ± 0.05 Post 58.53 ± 14.33 Rec 93.23 ± 13.06	↓[Na+] Pre 140.13 ± 1.64 Post 137.5 ± 42 Rec 139.5 ± 2.83 BDWL (−0.8 kg)	AKIN; Stage 1 (0, 0%) Stage 2 (2, 25%) EAH; Mil (3, 26.66%) Mod (1, 12.5%)
Jouffroy, 2019 [46]	21; -; 43 ± 7 yrs	80 km; 1; 1500+ m	-	↑SCR Pre 0.89 ± 16.1 Post 0.68.8 ± 21.4 ↓ eGFR Pre 90 ± 14 Post 119 ± 33	↓[Na+] Pre 143 ± 5 Post 141 ± 7 Rec(9 d) 141 ± 5 -	KDIGO; Risk (1, 6%) Injury (0, 0%)
Hoffmann, 2016 [47]	627; 187 F, 440 M; 41.36 yrs	161 km; 1; 5500+, 7000− m	-	↑SCR Group Risk 1.46 (1.39–1.61) Group Injury 1.98 (1.85–2.57)	-	RIFLE; Risk (227, 36.2%) Injury (31, 4.9%)
Pradas, 2021 [53]	20; 0 F, 20 M; 42.45 yrs	108 km; 1; 5800+ m	14.4 ± 4.4 °C; 57 ± 16.1%	↑SCR amateur Pre 90.1 ± 13.1 Post 120.8 ± 23.4 ↑SCR high lvl Pre 85 ± 10 Post 133 ± 11	↓[Na+] amateur Pre 140 ± 2 Post 140 ± 2 ↓[Na+] high lvl Pre 140 ± 1 Post 141 ± 3	RIFLE; Risk (0, 0%) Injury (0, 0%) EAH; Mil (0, 0%)
Khodaee, 2021 [54]	37; 8 F, 29 M; 43 yrs	161 km; 1; 2800+ m	2–22 °C; 57%	↑SCR Pre 0.99 ± 0.15 Post 68.8 ± 21.4	-	RIFLE Risk (18, 49%) Injury (0, 0%)
Le Goff, 2020 [48]	33; 0 F3 M; 45.8 ± 8.7 yrs	64.2 km; 1; 1400+ m	-	↑SCR Pre 9.14 ± 1.05, Post 12.46 ± 2.09 Rec (3 h) 11.03 ± 2.0	-	AKIN; Risk (0, 0%)
Lipman, 2016 [49]	128; 36 F, 92 M; 39.6 ± 9 yrs	250 km; 6; Nr	-	↑SCR Pre 0.72 ± 0.13 Post 1.28 ± 0.34	-	RIFLE; Risk (62, 48.4%) Injury (36, 28.1%)
M. Navarro, 2020 [50]	65 km race 4 F,15 M; 107 km race 17 F, 26 M 41 ± 6 yrs	107 km and 65 km; 1; 5604+, 4356− m	-	↑SCR ↑107 km >↑65 km Post, Post 24 h ↓ eGFR 65 km = 107 km Post, Post 24 h	↓[Na+] 65 km race BDWL (−6, +9) kg	RIFLE; Risk (32, 51.6%) Injury (0, 0%) EAH: Mil (3, 4.83%) Mod (2, 3.22%) Sev (0, 0%)
Belli, 2018 [51]	6; 0 F,6 M; 47 ± 5 yrs	217 kms; 1; 12,200 m;	-	↑SCR Pre 1.00 ± 0.03, Post 1.33 ± 0.08 ↓ eGFR Pre 89 ± 5 Post 65 ± 5	↓[Na+] Nr; BDWL (−4.1 ± 0.7%)	RIFLE Risk (0, 0%)
Cairns, 2016 [52]	15; 3 F,12 M; (40.7–50.6) yrs	100 km y 174 km; -	17.3–21.6 °C -	↓ eGFR EAH mod-31, −43 y-8 ↓ eGFR EAH mil −26,−25 y-7 ↓ eGFR No EAH −11,−20 y-1	-	RIFLE; Risk (0, 0%) EAH; Mil (3, 20%) Mod (7, 46%)
Boulter, 2011 [28]	4; 0 F,4 M; 35 ± 6 yrs	89 km; 1; -	14-(−24 °C); 63%	↑SCR Post (656–1139)	↓[Na+] Post (131–136) BDWL (−6 a +9) kg	RIFLE; Risk (0, 0%) Injury (4, 100%)

F: female; M: male; yrs: years; Km: kilometers; m: meters of elevation; ^+^: slope positive; −: slope negative; ±: positive and negative elevation mixed; AKI: acute kidney injury; EAH: exercise-associated hyponatremia; ↑SCR: elevation in creatinine serum from baseline; ↓eGFR: decrease in estimated glomerular filtrate; BDWL: body weight loss; -: not reported or not available; RIFLE and KDIGO AKIN criteria; EAH stages. Mil: mild; Mod: moderate; Sev: severe; Pre: baseline; Post: after race; Rec: recovery days after the race; ↓[Na+]: decrease in sodium serum concentration from baseline.

**Table 2 medicina-58-00569-t002:** Included studies examining EAH as the main target in subjects participating in ultra-trail races.

Author; Year; Reference	Population; Number; Sex; Age	Race; Length, Stages, Elevation	Temperature (Range); Humidity (% Relative)	AKI Biomarkers Pre, Post and Rec SCR (mg/dL) eGFR(mL/min/m^2^)	EAH Biomarkers [Na+] (mmol/L); Body Weight	AKI Criteria (Stage; *n*; %) EAH (Stage; *n*; %)
Page, 2007 [55]	123; 23 F, 97 M; -	60 km; 1; 1340+ m	8–14 °C; 60%	-	↓[Na+] (130–134); BDWL +1.32 kg	EAH; Mil (5, 4%)
Scotney, 2015 [27]	44; 8 F, 36 M; 39.5 yrs	82 km; 1; 1000+ m	6–15.6 °C; 79%	↑SCR Post 96.6 ± 20	↓[Na+] (132–147); BDWL 1.75 ± 1.36 kg (2,42–1.85%)	AKI; Risk (0, 0%) EAH Mod (2, 5%)
Chlíbkovál, 2019 [56]	20; 6 F, 14 M; 39.5 yrs	24 horas; 1; 764+ m	(−7.9)–20.6 °C; 88.3%	-	↓[Na+] (137–147); BDWL (−0.9%)	EAH Mil (0, 0%)
Hoffman, 2013 [57]	669; 229 F, 440 M 41.36 yrs	161 km; 1; 5500+, 7000− m	20–38 °C;	-	↓[Na+] (137–147); BDWL = 34.9% BDWL (0->3%) 46.6% BDWL (>3%)18.5%	EAH Mil (88, 13.2%) Mod (13, 1.9%)
Winger, 2013 [58]	207; 40 F, 167 M; 43.0 ± 9.6 yrs	161 km; 1; 5500+, 7000− m	-	-	↓[Na+] Post (131–134);	EAH; Mil (12, 5.8%)
Bracher, 2012 [59]	50; 0 F, 50 M; 47.8 yrs (45.4–50.3)	100 km; 1; 645+ m;	15.6–21.7 °C; 52–69%	↑SCR Pre 77.8 (74.5–81.1) Post 100.4 (93.3–107.5)	↓[Na+] Pre 136.6 (135.4–136.7) Post 100.4 (93.3–107.5)	AKI; Risk (0, 0%) EAH; Mil (0, 0%)
Knechtle, 2011 [60]	120; -; 44.5 ± 7 yrs	350 km; 7; 11,000+/− m	15.6–21 °C; 62%	-	↓[Na+] Pre 138.2 Post 138.49 BDWL (−0.2%)	EAH; Mil (7, 8%)
Hew-Butler, 2008 [61]	82; 24 F, 58 M; 43.7 ± 1.1 yrs	56 km; 1; 1000+ m	-	-	↓[Na+] Pre 139 ± 0.3 Post 138± 10.4 BDWL (−3.8%)	EAH, Mil (0, 0%)
Cuthill, 2009 [24]	4; 3 F,1 M; 41.5 yrs	152.88 km; 1; 4267 m	-	↑SCR Post 350.75 (114–761)	[Na+] Post 127.5 (120–134)	AKI; Injury (2, 50%) Failure (1, 25%) EAH; Mil (4, 100%)
Shepard, 2012 [62]	145; 0 F, 145 M;	100 km; 1; >500+ m	-	-	↓[Na+] Post (130–135)	EAH; Mil (7, 4.8%)
Costa, 2013 [63]	74; 28 F, 46 M; 41.8 yrs	225 km; 5; 2200+ m	32–40 °C; -	-	↓[Na+] - BDWL (−1.2%)	EAH; Mil (8, 42%)
Cejka, 2012 [64]	76; 0 F, 76 M; 47.1 yrs	100 km; 1; 645+ m	12–21 °C; -	-	[Na+] Pre 137.0 (2.7) Post 138.6 (2.6) BDWL (−1.8%)	EAH; Mil (4, 5.3%)
Knechtle, 2012 [65]	219; 219 M, 0 F; 46.2 ± 9.3 yrs	100 km; 1; 1050+ m	15–8 °C; -	-	[Na+] Pre 137.7 ± 2.3 Post 138.6 ± 2.7 BDWL	EAH; Mil (0, 0%)
Schenk, 2021 [66]	69 km race 0 F,11 M; 121 km race 0 F, 7 M 41.2 yrs	121 km and 69 km; 1; 7554+ m, 4260+ m	22–30 °C; -	-	↓[Na+] 69 km Pre 142.9 ± 1.9 Post 143.7 ± 2.1 ↓[Na+] 121 km Pre 142.0 ± 1.7 Post 142.6 ± 4.2 BDWL 69 km −3.1%; 121 km −2.7%	EAH 69 km Mil (0, 0%) Mod (0, 0%) EAH 121 km Mil (0, 0%) Mod (0, 0%)
Khodaee, 2021 [67]	84; 69 M, 15 F 42.1 ± 9 yrs	161 km; 1; 2800+ m	2–22 °C; 57%		↓[Na+] Pre 138.4 ± 1.7 Post 135.8 ± 3	EAH: Mil (17, 20%) Mod (1, 1.19%)

F: female; M: male; yrs: years; km: kilometers; m: meters of elevation; +: slope positive; −: slope negative; +/−: positive and negative elevation mixed; AKI: acute kidney injury; EAH: exercise-associated hyponatremia; ↑SCR: elevation in creatinine serum from baseline; ↓eGFR: decrease in estimated glomerular filtrate; BDWL: body weight loss; -: not reported or not available; RIFLE and KDIGO criteria; AKIN criteria; EAH stages. Thousand: mild; Mod: moderate; Sev: severe; Pre: baseline; Post: after race; Rec: recovery days after the race; ↓[Na+]: decrease in sodium serum concentration from baseline.

**Table 3 medicina-58-00569-t003:** AKI cases sorted by race category.

Race Category (Number of Studies)	Length (km)	Elevation (m)	Population; Gender; Age	AKI Cases (*n* and %)			
				Risk	Injury	Failure	Total
Medium (3)	65.33 ± 1.52	3630 ± 1950	60 (4 F,56 M) 42.98 ± 2.45	34 56.67%	0 0%	0 0%	34 53.67%
Large (3)	83.66 ± 4.75	1166 ± 288	69 (8 F,61 M) 39.60 ± 4.05	1 1.44%	4 5.80%	0 0%	5 7.24%
Extra (10)	136.22 ± 39.59	4851 ± 3440	826 (219 F, 607 M,) 42.92 ± 3.38	287 34.74%	34 4.11%	1 0.12%	304 38.98%
Multistage (2)	2500 ± 0	2000 ± 0	158 (43 F,115 M) 39.6 ± 2.1	70 44.30%	37 23.41%	0 0%	108 68.35%
Total (18)	127.82 ± 61.56	3560 ± 2951	1113 (274 F, 839 M)	392 35.22%	75 6.73%	1 0.1%	468 42.04%

**Table 4 medicina-58-00569-t004:** EAH cases sorted by the race category.

Race Category (Number of Studies)	Length (km)	Elevation (m)	Population; Gender, Age	EAH Cases (*n* and %)			
				Mild	Moderate	Severe	Total
Medium (7)	61 ± 8.30	2501 ± 1974	482 (422 F, 60 M) 42.98 ± 2.45	11 2.28%	2 0.41%	0 0%	13 2.69%
Large (3)	83.66 ± 4.75	1166 ± 288	69 (8 F, 61 M) 39.06 ± 4.05	4 5.79%	2 2.90%	0 0%	6 8.69%
Extra (11)	124.63 ± 27.82	3746 ± 2613	1320 (307 F, 1013 M) 42.68 ± 2.95	142 10.75%	19 1.43%	0 0%	161 12.19%
Multistage (2)	287.50 ± 88.38	7067 ± 5562	194 (52 F, 142 M) 43.15 ± 1.90	15 7.73%	0 0%	0 0%	15 7.73%
Total (19)	114.08 ± 67.38	3319 ± 2792	2065 (427 F, 1638 M) 42.1 ± 1.90	172 7.73%	23 1.11%	0 0%	195 9.44%

**Table 5 medicina-58-00569-t005:** AKI+EAH cases sorted by race category.

Race Category (Number of Studies)	Length (km)	Elevation (m)	Population; Gender; Age	AKI+EAH (*n* and %)
Medium (2)	66 ± 1.41	4750 ± 353	27 (4 F, 23 M) 41.55 ± 0.07	5 18.51%
Large (3)	83.7 ± 4.72	1166 ± 288.67	69 (8 F, 61 M) 39.60 ± 4.05	4 5.79%
Extra (5)	113 ± 21.90	3342 ± 2366	132 (23 F, 109 M) 43.59 ± 2.47	18 13.63%
Total (10)	95 ± 25.2	2971 ± 2094	228 (35 F, 193 M) 41.99 ± 3.12	36 11.84%

**Table 6 medicina-58-00569-t006:** Methodological quality assessment of included studies using two assessment tools.

Study	Quality Assessment Tool ^1^ [41]															Level of Evidence ^2^
	C1	C2	C3	C4	C5	C6	C7	C8	C9	C10	C11	C12	C13	C14	Q	[42]
Hoffman, 2013 [57]	yes	yes	no	no	yes	yes	yes	yes	yes	yes	yes	na	na	no	Good	A
Winger, 2013 [58]	yes	yes	yes	nr	no	no	yes	yes	no	no	yes	na	no	no	Good	A
Bracher, 2012 [59]	yes	yes	yes	yes	no	yes	yes	na	yes	no	yes	na	yes	yes	Good	A
Knechtle, 2011 [60]	yes	yes	nr	na	yes	yes	yes	yes	yes	no	yes	nr	nr	yes	Good	A
Hew-Butler, 2008 [61]	yes	yes	nr	nr	yes	yes	yes	yes	yes	no	yes	na	yes	yes	Good	A
Lipman, 2014 [43]	yes	yes	yes	no	no	yes	yes	yes	yes	no	yes	na	yes	na	Fair	B
Poussel, 2020 [44]	yes	yes	yes	yes	yes	yes	yes	no	yes	no	yes	na	no	na	Fair	B
Hoppel, 2019 [45]	yes	yes	no	yes	no	yes	yes	na	yes	yes	yes	na	no	na	Fair	B
Jouffroy, 2019 [46]	yes	yes	yes	yes	no	yes	yes	na	yes	yes	yes	na	yes	yes	Fair	B
Hoffmann, 2015 [47]	yes	yes	no	no	no	no	no	yes	yes	no	yes	na	no	yes	Fair	B
Le Goff, 2015 [48]	yes	no	no	yes	no	yes	yes	na	yes	no	yes	na	nr	yes	Fair	B
Lipman, 2016 [49]	yes	yes	yes	no	no	yes	yes	yes	yes	no	yes	na	yes	na	Fair	B
M. Navarro, 2020 [50]	yes	yes	yes	na	yes	yes	yes	na	yes	yes	yes	na	yes	yes	Fair	B
Khodaee, 2021 [54]	yes	yes	yes	no	no	yes	yes	yes	yes	no	yes	na	yes	na	Fair	B
Khodaee, 2021 [67]	yes	yes	yes	no	no	yes	yes	yes	yes	no	yes	na	yes	na	Fair	B
Pradas, 2021 [53]	yes	yes	yes	yes	yes	yes	yes	no	yes	no	yes	na	no	na	Fair	B
Schenk, 2021 [66]	yes	yes	yes	yes	yes	yes	yes	no	yes	no	yes	na	no	na	Fair	B
Belli, 2018 [51]	yes	yes	no	yes	yes	yes	yes	na	yes	yes	yes	na	yes	yes	Fair	B
Cairns, 2016 [52]	yes	yes	yes	yes	no	no	yes	yes	yes	yes	yes	na	no	yes	Fair	B
Boulter, 2011 [28]	yes	yes	nr	no	na	na	yes	na	yes	na	yes	na	na	yes	Fair	B
Page, 2007 [55]	yes	yes	nr	yes	no	yes	yes	yes	yes	no	yes	na	nr	yes	Fair	B
Scotney, 2015 [27]	yes	yes	yes	yes	yes	no	yes	yes	yes	no	yes	na	yes	yes	Fair	B
Chlíbková1, 2019 [56]	yes	yes	yes	no	yes	yes	yes	no	yes	no	yes	na	no	yes	Fair	B
Cuthill, 2009 [24]	yes	yes	na	yes	na	na	yes	yes	yes	no	yes	na	na	yes	Poor	C
Shepard, 2012 [62]	yes	nr	nr	no	yes	yes	yes	no	nr	no	yes	na	nr	no	Poor	C
Costa, 2013 [63]	yes	yes	nr	no	no	yes	yes	yes	yes	yes	yes	na	yes	yes	Poor	C
Cejka, 2012 [64]	yes	yes	yes	nr	no	yes	yes	yes	yes	no	yes	na	nr	yes	Poor	C
Knechtle, 2012 [65]	yes	yes	nr	yes	no	yes	yes	yes	yes	no	yes	nr	yes	yes	Poor	C

^1^ Level of Evidence; ^2^ Quality Assessment Tool for Observational Cohort and Cross-Sectional Studies; nr, not reported; na, not applicable; cd, cannot be determined, C1–C14, check list criteria; Q, quality.

## Data Availability

Not applicable.

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
