# Peer review of "Acute Kidney Injury and Hyponatremia in Ultra-Trail Racing: A Systematic Review"

_medicina, 2022, doi:10.3390/medicina58050569_

Round 1

Reviewer 1 Report

Very detailed and well-crafted review dealing with useful issues, newer concepts in exercise-associated hyponatremia: an association between rhaddomyolysis and EAH.

Author Response

Dear reviewer,

We appreciate your review.  We feel proud that you have found it worth reviewing our systematic review.  It has required considerable effort to summarize outcomes and variables related to AKI and EAH in these races, but we think this approach offers new data to trainers and especially medical staff to prevent runners from suffering AKI and EAH.

Despite the amount of research conducted in the last fifteen years, there is still no consensus on how to prevent the appearance of these medical conditions. In our humble opinion, this systematic review may help runners and medical staff apply running strategies to avoid the concurrence of these pathologies.

Yours faithfully,

Carlos Castellar Otin

Reviewer 2 Report

Introduction:

As EAH is largely an acute hyponatremia, the definition of EAH must include that it is an acute process. Although hypervolemic hyponatremia is the main cause of EAH, recent evidence support hypovolemic hyponatremia also can lead to EAH. The author must include this pathogenesis.

The statement "AKI is defined as an abrupt (within hours) decrease in kidney function, which encompasses both injury and impairment" is too simple. Because three AKI classification was used in your review, the author must introduce RIFLE, AKIN and KDIGO classification.

Discussion:

Only one case met failure stage of AKI. What type of the race and the case’s outcome ?

Reference:

Several references are in wrong number, for example the correct number of reference 46 is 45, the correct number of reference 54 is 53, the correct number of reference 78 is 77. The author must check ALL references again to provide correct number.

Other comment

Line 52: The impact of such races "on the on the" human body has been investigated from a broad perspective. Is there a duplication ?

Lin 72: EAH and Acute kidney injury (AKI), A must be lowercase.

Line 82: "Effect of prolonged heavy exercise on renal function and urinary protein excretion." What is the sentence mean ?

Line 163: "low" chart -> flow chart

Lin 267: 7,24% -> 7.24%

Line 324: There is no risk stage in KDIGO classification, do you mean RIFLE classification ?

Line 354: Body weight and excessive consumption of beverages with or without [Na+] supplementation are two relevant factors to develop EAH "to EAH". Is there a duplication ?

Author Response

Dear reviewer,

Thank you for allowing us to submit a revised draft of our manuscript titled “Acute Kidney Injury and Hyponatremia in Ultra-Trail Racing: A Systematic Review.” to "Medicina".

We appreciate the time and effort that you have dedicated to providing your valuable feedback on our manuscript. Consequently, we have been able to incorporate changes to reflect most of the comments provided by you. We have highlighted the changes within the manuscript in green colour and a comment has been added in the original document.

Here is a point-by-point response to your main notes and concerns. Additionally following the manuscript that you gently attached in the review process we have corrected minor issues as misspelling or format corrections.

  • Comment 1 "As EAH is largely an acute hyponatremia, the definition of EAH must include that it is an acute process. Although hypervolemic hyponatremia is the main cause of EAH, recent evidence support hypovolemic hyponatremia also can lead to EAH. The author must include this pathogenesis. "
  • Response 1: We agree. We have added the aetiology mechanism that you suggested. We have referenced it and additionally we have commented euvolic aetiology too.   
  • Comment 2 "AKI is defined as an abrupt (within hours) decrease in kidney function, which encompasses both injury and impairment" is too simple. Because three AKI classification was used in your review, the author must introduce RIFLE, AKIN and KDIGO classification. "
  • Response: We have rewritten the definition of AKI. Equally, we have added the differences among the three criteria. New references have been added to support properly these definitions.
  • Comment 4. "Only one case met failure stage of AKI. What type of the race and the case’s outcome? "
  • Response: We have specified the type of race and the treatment required by the patient and the prognosis.
  • Comment 5

“Several references are in wrong number”

  • Response: Due to some problems with the template format, some references were disordered. We have revised and sorted all references. Additionally, we have updated the systematic review to meet validity PRISMA criteria, so new articles and references have been added.
  • Other comments. Minor issues related to duplication and some misspelling
  • Response: All minor mistakes have been rewritten and solved following your suggestions.

We hope you may find convenient the information added in this email, and please do not hesitate to contact us regarding any queries you might have.

Yours faithfully,

Carlos Castellar Otín.

Reviewer 3 Report

Manuscript ID: medicina-1644661

The authors present a comprehensive summary of their review of “Acute Kidney Injury and Hyponatremia in Ultra-Trail Racing: A Systematic Review”. They provided evidence that the extra and medium races showed a similar incidence of AKI+EAH. This review is quite a bit interest. However, some of main important issues need to be verified to improve their work as following

            Major concerns.

  • This review was absent of PRISMA checklist. In addition, the protocol registration was absent. Prospective registration of systematic reviews promotes transparency, helps reduce potential for bias and serves to avoid unintended duplication of reviews.
  • How were the search terms defined? Is there a pre-test to define the search strategy used in each database? Did the search strategy the same in all databases? Search terms in PubMed and the others are different. The authors should attach syntax used in each database in supplementary.
  • Keywords: Considering the low number of search yields from most databases (e.g., Pubmed n=10), I doubt the appropriateness of the search terms as it seems the authors have used. This leads to the concern of the search strategies performance to collect all relevant studies. I wonder if the search strategies have been developed with the help of a librarian or experienced reviewers in the field. Generally, if the search yield is too low, systematic reviewers would need to modify the search term to ensure that it well cover most of the related papers. This could be done by in the PICO such as Control and Outcome; employ a free text rather than thesaurus search terms. I think the keywords for the outcome domain were not comprehensive enough to capture all potential synonyms of the outcomes of interested in this study, and therefore would be best not to limit the search with these terms. An example of the search strategies used for a particular database would show a transparency in this step (I don’t think the keywords presented is sufficient as an example of a search strategies as recommended in PRISMA).
  • Please add more information of source of heterogeneity in discussion part.
  • How authors dealt with missing data. Did you receive all answers from authors of the studies or make some imputation? In general, if the study did not report the data of the primary or secondary outcomes measures, the authors should contact via email to provide this information. Have you considered to contact authors via researchers’ network ResearchGate (https://www.researchgate.net/), Academia (https://www.academia.edu/), Loop (https://loop.frontiersin.org/) or Quora (https://es.quora.com/)? Nowadays these platforms are very useful and efficient canals to contact authors.
  • In "Methods" part, the independent investigators who extracted data and evaluated quality of each study should be clarify.
  • The publication bias evaluation is missing. The authors must demonstrate both statistics and visualization. In addition, I suggest plot the funnel and contour-enhanced funnel in the graphic and not only the studies for better interpretation. Besides, it is necessary to present the p value for this analysis. Please add this information in supplementary.
  • The date of the literature was January 2021. In accordance with guidelines, the literature search should be performed until six months before the submission of the manuscript for publication. If the manuscript is accepted for publication, it is already outdated.

Author Response

Dear reviewer,

Thank you for allowing us to submit a revised draft of our manuscript titled “Acute Kidney Injury and Hyponatremia in Ultra-Trail Racing: A Systematic Review.” to "Medicina".

We appreciate the time and effort that you have dedicated to providing your valuable feedback on our manuscript. In consequence, we have been able to incorporate changes to reflect most of the comments provided by you. We have also highlighted in fuchsia colour the changes within the manuscript to facilitate the location of them.

Here is a point-by-point response to your main notes and concerns. Additionally following the manuscript that you gently attached in the review process we have corrected minor issues as misspelling or format corrections.

  • Comment 1This review was absent of PRISMA checklist. In addition, the protocol registration was absent. Prospective registration of systematic reviews promotes transparency, helps reduce potential for bias and serves to avoid unintended duplication of reviews.”
  • Response 1: The documents requested have been added. According to PRISMA, it should not be included as part of the main manuscript. Therefore, we decided it should be uploaded as a separate file only if the editors or reviewers requested it. Once the file has been asked, it has been attached along with the other files in the submission menu (Moher et al., 2009)

The systematic review was submitted to PROSPERO on the 7th of March (we include proof of submission). However, it has not been registered yet due to COVID -19 and the vast number of registers requesting PROSPERO registration. The expected date is the 7th of April. No sooner has the manuscript been accepted, we shall provide the proof of registration to the editor as part of the submission process. We, as investigators, support certainly the essential role that PROSPERO plays in the investigation process. Accordingly, to our beliefs, before starting the writing process, PROSPERO was consulted to check the existence of any review with the same PICO or main objectives. So far, this is the first review to clarify the concurrence of these two medical conditions in these kinds of races.

  • Comment 2:” How were the search terms defined? Is there a pre-test to define the search strategy used in each database? Did the search strategy the same in all databases? Search terms in PubMed and the others are different. The authors should attach syntax used in each database in supplementary.”
  • Response 2: Agree. We have described all the terms and words used in the search. Following your comments, the different equations used in every database are in Appendix A. We use the same system in every single database where we conducted the search. To ensure the validity of the process we used four different high-quality databases related to medicine and sports. The low number of registers found depends solely on the field of study or contents of every one of them. The reviewer should consider the heterogeneity present in the ultra-trail races along with the different criteria used to diagnose AKI. In order to ensure the validity of the review Mesh term related to AKI and free words about ultra-trail races were defined.
  • Comment 3: “Keywords: Considering the low number of search yields from most databases (e.g., Pubmed n=10), I doubt the appropriateness of the search terms as it seems the authors have used. This leads to the concern of the search strategies performance to collect all relevant studies. I wonder if the search strategies have been developed with the help of a librarian or experienced reviewers in the field. Generally, if the search yield is too low, systematic reviewers would need to modify the search term to ensure that it well cover most of the related papers. This could be done by in the PICO such as Control and Outcome; employ a free text rather than thesaurus search terms. I think the keywords for the outcome domain were not comprehensive enough to capture all potential synonyms of the outcomes of interested in this study, and therefore would be best not to limit the search with these terms. An example of the search strategies used for a particular database would show a transparency in this step (I don’t think the keywords presented is sufficient as an example of a search strategies as recommended in PRISMA).”
  • Response 3: We fully agree. We have redefined the words related to AKI using MeSH terms. For ultra-trail races, we included additional free text words to find the most relevant articles for the main objective of our study (i.e., longer than the marathon and with a positive or negative elevation higher than 500 m and in off-road settings). As a consequence, a new flow diagram resulted in more coherence and, subsequently, the number of potential studies increased.
  • Comment 4: “Please add more information of source of heterogeneity in discussion part.”
  • Response 4: We have added a specific point to tackle this issue. (Limitations). The design of the studies (cross-sectional) and the different biomarkers for acute kidney damage are the principal sources of heterogeneity. In addition, the high number of definitions and terms to term these races hinder the selection of keywords. A whole paragraph focusses on discussing the quality of the criteria and diagnosing methods for AKI in this sport.
  • Comment 5: “How authors dealt with missing data. Did you receive all answers from authors of the studies or make some imputation? In general, if the study did not report the data of the primary or secondary outcomes measures, the authors should contact via email to provide this information. Have you considered to contact authors via researchers’ network ResearchGate (https://www.researchgate.net/), Academia (https://www.academia.edu/), Loop (https://loop.frontiersin.org/) or Quora (https://es.quora.com/)? Nowadays these platforms are very useful and efficient canals to contact authors.”
  • Response 5: We included all studies which meet the inclusion criteria. Additionally, to ensure the quality and to reduce bias selection of the systematic review we used two high-quality tools specifically designed for cross-sectional studies. Studies lacking vital data related to the primary outcomes (biomarkers pre and post or AKI and EAH incidence) were not included. Tables 1 and 2 depict all the characteristics of every study included. We did not make any imputation. We appreciate the idea that you have suggested and we would like to use it for future investigations. This time, we could not implement this idea because of the lack of time. Systematic reviews require a fast publication process in order to ensure its validity, recollecting all missing data using the different apps suggested by you would have implied a longer process.
  • Comment 5:In "Methods" part, the independent investigators who extracted data and evaluated quality of each study should be clarify.”
  • Response 5: We agree. We have added the names and the whole process for the extraction data. Equally, the assessment of the quality of the studies have been explained, as well as the authors who performed every part of the process.
  • Comment 6: The publication bias evaluation is missing. The authors must demonstrate both statistics and visualization. In addition, I suggest plot the funnel and contour-enhanced funnel in the graphic and not only the studies for better interpretation. Besides, it is necessary to present the p value for this analysis. Please add this information in supplementary.
  • Response 6: The main objective of the present systematic review is to assess the AKI and EAH cases together or alone in ultra-trail racing. Therefore, a descriptive analysis is, to our belief, this design is relevant enough for the objective of this review. Previous reviews evaluating AKI or EAH in endurance sports have not included this meta-analysis. We agree, by adding a meta-analysis the study would be more precise. But we believe that this analysis meets the objective of the study perfectly.

The graphics that you suggested would be convenient but to our belief, the tables nº 3, 4 and 5 are clear enough and summarizes perfectly the descriptive analysis conducted facilitating to potential readers the incidence of AKI and EAH and the distribution of cases according to the race category.

  • Comment 7: “The date of the literature was January 2021. In accordance with guidelines, the literature search should be performed until six months before the submission of the manuscript for publication. If the manuscript is accepted for publication, it is already outdated.”
  • Response 7: The systematic review is updated with a new deadline of December 31st 2021. Accordingly, this systematic review contains the latest articles offering valuable data related to AKI, Hyponatremia and ultra-trail races. It is the first review to show AKI and EAH cases according to the type of race. This data is vital for medical staff to avoid the appearance of these medical conditions and, consequently, to ensure runners' health and wellness.

We hope you may find convenient the information added in this email, and please do not hesitate to contact us regarding any queries you might have.

Yours faithfully,

Carlos Castellar Otín
